# Effects of Methyl Jasmonate and Nano-Methyl Jasmonate Treatments on Monastrell Wine Volatile Composition

**DOI:** 10.3390/molecules27092878

**Published:** 2022-04-30

**Authors:** María José Giménez-Bañón, Juan Daniel Moreno-Olivares, Diego Fernando Paladines-Quezada, Juan Antonio Bleda-Sánchez, José Ignacio Fernández-Fernández, Belén Parra-Torrejón, José Manuel Delgado-López, Rocío Gil-Muñoz

**Affiliations:** 1Murcian Institute of Agricultural and Environment Research and Development (IMIDA), Ctra. La Alberca s/n, 30150 Murcia, Spain; gimenezba@hotmail.com (M.J.G.-B.); juand.moreno5@carm.es (J.D.M.-O.); diegopaladines@hotmail.com (D.F.P.-Q.); juanantonio.bleda@carm.es (J.A.B.-S.); josei.fernandez@carm.es (J.I.F.-F.); 2Department of Inorganic Chemistry, Faculty of Science, University of Granada, 18071 Granada, Spain; belenparrato@ugr.es (B.P.-T.); jmdl@ugr.es (J.M.D.-L.)

**Keywords:** nanoparticles, elicitor, aroma, foliar application, sensorial analysis

## Abstract

The application of methyl jasmonate (MeJ) as an elicitor to enhance secondary metabolites in grapes and wines has been studied, but there is little information about its use in conjunction with nanotechnology and no information about its effects on wine volatile compounds. This led us to study the impact of nanoparticles doped with MeJ (Nano-MeJ, 1mM MeJ) on the volatile composition of Monastrell wines over three seasons, compared with the application of MeJ in a conventional way (10 mM MeJ). The results showed how both treatments enhanced fruity esters in wines regardless of the vintage year, although the increase was more evident when grapes were less ripe. These treatments also achieved these results in 2019 in the cases of 1-propanol, ß-phenyl-ethanol, and methionol, in 2020 in the cases of hexanol and methionol, and in 2021, but only in the case of hexanol. On the other hand, MeJ treatment also increased the terpene fraction, whereas Nano-MeJ, at the applied concentration, did not increase it in any of the seasons. In summary, although not all families of volatile compounds were increased by Nano-MeJ, the Nano-MeJ treatment generally increased the volatile composition to an extent similar to that obtained with MeJ used in a conventional way, but at a 10 times lower dose. Therefore, the use of nanotechnology could be a good option for improving the quality of wines from an aromatic point of view, while reducing the necessary dosage of agrochemicals, in line with more sustainable agricultural practices.

## 1. Introduction

The aromatic profile is an important factor in wine quality. The origin of the different aromas is complex and includes free aromas in the grapes (monoterpenes, norisoprenoids, aliphatics, phenylpropanoids, methoxypyrazines, and volatile sulfur compounds), a pool of precursors, including glycosides of aroma molecules (glutathionyl and cysteinyl precursors, dimethyl sulfide precursors, mainly S-methylmethionine), amino acid, and fatty acid precursors [1]. The origin of the different aromas also includes the products of the fermentation process, including higher alcohols, volatile fatty acids, and esters, and aromas from oxidative processes produced during wine storage [2]. It is important to note that many factors can affect the development of aromas and their precursors in grapes, including agronomic practices [3], oenological techniques [4], weather conditions [5], and grape variety [1].

*Vitis vinifera* L. cv Monastrell is mainly cultivated in the Murcia and Alicante regions of southeastern Spain, where it is perfectly adapted to poor soil and extreme weather conditions with few rainy periods [6]. From an aromatic point of view, the Monastrell grape is a neutral variety with a poor monoterpene content, meaning that the most important flavor compounds found in the wine arise from the fermentation process [7]. In addition, in neutral grapes such as Monastrell, the degree of ripeness influences the profiles of the fermentative compounds [8].

Many agronomic practices, such us leaf removal, canopy training systems, foliar fertilization, irrigation, and exogenous product applications (so-called elicitors) can affect the primary and secondary metabolism, sometimes modifying the aromatic profile [3]. In attempts to improve wine aroma quality, numerous studies have investigated the effects of different elicitors on the volatile composition of grapes, finding that their impact varies, depending on several factors such as grape cultivar, type of elicitor, and dose. Among these compounds is the jasmonate family, which includes jasmonic acid and its ester methyl jasmonate (MeJ). Both compounds are extensive in the plant kingdom and their induction of the defensive response in plants has been used in several crops, increasing secondary metabolites [9,10]. In viticulture, too, good results have been obtained, including improved phenolic composition [11,12,13], the enhanced concentration of some amino acids [14], and a better aromatic profile [15,16,17].

In spite of the good results obtained when MeJ is applied in vineyards, its high cost, low water solubility, low thermal stability, and phytotoxicity limit its efficient applicability, particularly at high concentrations [18]. An interesting alternative to overcome these limitations is the use of nanotechnology. The benefits of nanotechnology have already been demonstrated in several fields, medicine being the most representative. More recently, nanotechnology has been integrated into agriculture with very promising results [19,20,21,22]. Among existing nanoparticles, calcium phosphate nanoparticles (CaP), which mimic the inorganic phase of bone [23], are of special interest in agriculture because they are biodegradable and biocompatible materials with tunable solubility and high specific area, which allows the loading of active species, such as nutrients or elicitors [24]. However, the use of CaP nanoparticles in viticulture is still in its infancy. The efficiency of MeJ-doped nanoparticles in producing stilbenes [25] and increasing the amino acid composition of Monastrell grapes [26] and wines has been reported in an experiment covering two seasons. However, the volatile composition of wine resulting from Nano-MeJ treatments to Monastrell grapes has not been studied.

This contribution aims to provide information on the effect of elicitation by MeJ used in a conventional manner and MeJ supported by calcium phosphate nanoparticles (Nano-MeJ). The resulting wine volatile composition are compared, with the aim of establishing a more sustainable way to improve wine quality.

## 2. Results

### 2.1. Oenological Must and Wine Parameters

The oenological parameters of musts and wines of untreated and treated grapes over three consecutive seasons are shown in Table 1.

°Brix, total acidity, pH, tartaric acid, and malic acid were analyzed in the musts. In general terms, although we did not find differences for the °Brix between the control grapes and those treated in the same year, the lowest °Brix corresponded to the first year of study and the highest value to the 2020 season. This parameter is important, because sugar as a primary metabolite also influences several secondary metabolites, especially the concentrations of aroma compounds [27,28]. Moreover, acidity and the malic acid content were higher in musts from MeJ-treated grapes.

With regard to the different seasons, in general terms, neither of the MeJ treatments (conventional or nanoparticles) affected the must oenological parameters in 2019. These findings are similar to those reported by Gutiérrez-Gamboa et al. [29] for the 2014 season in Tempranillo grapes, in that MeJ did not affect the oenological parameters of the must, and Portu et al. in Tempranillo grapes for the 2013 [30] and 2015 [12] seasons. However, in our study, total acidity and malic acid were significantly affected in 2020 and 2021 when grapes were treated with MeJ, but not when nanoparticles were applied. These results agree with the findings of D’Onofrio et al. [31], who found that MeJ treatment delayed technological maturity. Finally, in 2021, the pH was lower in MeJ-treated grapes but not in grapes when Nano-MeJ was applied. 

Other authors have found that must parameters may differ when grapes are treated with MeJ. For example, Ruiz-García et al. [13] observed an increase in total acidity and tartaric acid and a decrease in °Brix in Monastrell grapes in the 2009 season, whereas in 2010 the pH, °Brix, and malic acid concentration increased. Further, Paladines-Quezada et al. [32] studied the effect of foliar application of MeJ and benzothiadiazole (BTH) during two vintages and found that only the pH increased in control grapes in the 2017 season.

This diversity in the results found between seasons might be explained by differences in climate conditions. As can be seen in Figure 1A–C, the driest and warmest season was 2020, with a total rainfall of 366 Lm*^−^*^2^ and an average temperature of 15.5 °C, while in 2019 rainfall was 438 Lm*^−^*^2^ and the average temperature 14.8 °C. The corresponding figures for 2021 were 425 Lm*^−^*^2^ and 15.2 °C. As can be observed in Figure 1A, precipitation mainly occurred at three times, the amount of rain that fell in September (150 Lm*^−^*^2^), just before the harvest, being of particular note, in contrast to Figure 1B and 1C, which shows a more homogeneous distribution. These weather conditions provoked an early harvest in 2019 because of torrential rains in mid-September, and consequently the grapes were harvested with a lower °Brix.

The following oenological parameters were analyzed: alcohol %, volatile acidity, total acidity, pH, and malic acid. Alcohol percentage was not affected by the treatments in any season, while volatile acidity progressively decreased in the three seasons in the case of the Nano-MeJ treatment, although not significantly in 2019. Total acidity and malic acid showed a similar behavior pattern, with total acidity levels decreasing and malic acid increasing in grapes treated with MeJ and Nano-MeJ, although not always to a statistically significant degree. Other authors, such as Portu et al. [28], observed no differences between the control and MeJ treatment for Tempranillo in the 2013 season, although the same author found that pH fell in the MeJ treatment in the 2014 season [33].

### 2.2. Effect of MeJ and Nano-MeJ Treatments on Wine Volatile Composition at the end of Alcoholic Fermentation

Table 2 shows the effect of MeJ treatments on wine volatile composition (fatty acids, alcohols, esters, terpenes, and methionol) at the end of alcoholic fermentation over three consecutive seasons (2019, 2020, and 2021). In general terms, the concentration of volatile compounds in wines made from both treated and untreated grapes were lower in 2019, perhaps due to the climatic conditions (high rainfall just before harvest) having a dilution effect on the compounds.

#### 2.2.1. Fatty Acids

Fatty acids are formed mainly by yeasts from the biosynthesis of acetyl-CoA during alcoholic fermentation. The different groups of fatty acids produced are short-chain fatty acids (<6 carbons), medium-chain saturated fatty acids (6–12 carbons), long-chain saturated fatty acids (>12 carbons), and long-chain unsaturated fatty acids. The volatile fraction corresponding to short and medium-chain fatty acids [2,34] and medium-chain fatty acids contribute negative cheese notes to wine aroma [35].

Substantial differences were observed between the three seasons in this group of compounds. In 2019, only one fatty acid was found (octanoic acid) and no significant difference between treatments was detected. On the other hand, in 2020, the contribution of fatty acids (hexanoic and octanoic) was greater, hexanoic acid only being significantly higher in the MeJ treatment, but not when nanoparticles were applied. In 2021, the contribution of fatty acids was again of note in both treatments and very similar to that found for the previous season. However, on this occasion, the increase was more evident, the differences being statistically significant with respect to the control wines for both treatments. Gomez Plaza et al. [36] also detected decanoic acid in Monastrell wine in 2010, but no significant differences were found in wines made from grapes treated with MeJ and wines made from the untreated grapes, while Ruiz-García et al. [16] in 2011 found an increase in hexanoic acid in wines from Monastrell grapes treated with MeJ.

#### 2.2.2. Higher Alcohols

Higher alcohols were the most abundant volatile compounds detected in wines in the three seasons studied, although some differences existed between them for the different compounds analyzed. Similarly, other authors reported that these compounds were the most abundant family of volatile compounds in Tempranillo blanco wines [37,38].

Higher alcohols are volatile compounds derived from yeast amino acid and sugar metabolism [39]. Among the principal descriptors of these alcohols is pungency, so a high alcohol value would be considered a deficient aroma. Etievant [40] reported that total concentrations of around 300 mg L*^−^*^1^ could be regarded as positive for quality wines, increasing their flowery and fruity complexity, especially β-phenyl-ethanol with its rose notes [41], whereas more than 400 mg L*^−^*^1^ would provide negative notes unfavorable for the final quality of the wines.

In 2019, both MeJ and Nano-MeJ increased the total content of higher alcohols to the same extent (19%) although the concentration of Nano-MeJ used was ten times lower. However, of the different analyzed, this increase was only significant in the case of 1-propanol and β-phenyl–ethanol for both treatments. Other compounds, such as 2-methyl-1-propanol, 2-methyl-1-butanol, and 3-methyl-1-butanol, also increased in concentration in both treatments (14%, 19%, and 19%, respectively, compared with the control wine), although the increases were not statistically significant. Finally, C6 alcohols (1-hexanol and E-3-hexen-1-ol), which are responsible for green notes, increased by 11% and 17%, respectively, in wines made from MeJ-treated grapes but not in wines from the Nano-MeJ-treated grapes, although, again, this increase was not statistically significant. This could be explained because the concentration of C6 compounds is higher during pre-veraison and veraison, but then starts to decline until harvest [42].

In contrast, in 2020, we observed a slight and not statistically significant decrease in total high alcohol content in wines elaborated with treated grapes. Only the 1-hexanol content increased in both treatments (8% in MeJ and 10% in Nano-MeJ), perhaps because the grapes were harvested in a more mature state than in the other two years of the study. Zhao et al. [43] showed how sequential harvesting treatments in Cabernet Sauvignon grapes produced significantly higher alcohol levels due to the presence of isopentanol and phenyl-ethyl alcohol, while in the rest of the alcohol compounds this increase was not evident, as was the case in our study.

In 2021, the total amount of higher alcohols increased by 12% for both treatments, but only 1-hexanol increased to a statistically significant extent in the MeJ treatment. Among the authors who describe wines made from grapes treated with MeJ, D´Onofrio et al. [30] found that some alcohols, including β-phenyl-ethanol, increased in Sangiovese wine in 2015, while Gomez-Plaza et al. [36] described an increase in total higher alcohols in Monastrell wines in 2010.

Finally, of note was the fact that the percentage of C6 alcohols with respect to the total alcohol content was higher in 2019 than in 2020 and in 2021, perhaps due to the lower maturity of grapes in 2019; several authors also mentioned a decrease in C6 alcohols compared with other alcohols as grape maturity increased [44].

#### 2.2.3. Esters

Esters are mostly produced by yeasts via lipid and acetyl-CoA metabolism during fermentation, although some may come from the esterification of alcohols and acids during wine ageing [30]. Esters are the most important odorants in wines, since they impart abundant floral and tropical fruity aromas [45].

Two main groups of esters can be differentiated for their sensory contribution: ethyl acetate esters (acetic vector) and the rest of the esters (fruity vector). Ethyl acetate was the main ester detected in our control wine and in the wines elaborated with treated grapes during the three studied seasons, although no statistical differences were found among them. Regarding the fruity vector, all the treatments increased this parameter in all years. The greatest increase was found in 2019 (100% and 75% increase for the MeJ and Nano-MeJ treatments, respectively), followed by increases of 73% and 62% in 2020, and 35% and 20% in 2021. It is important to underline that these results were obtained using doses of MeJ ten times lower in Nano-MeJ-treated grapes compared with the MeJ treatment. In addition, the results suggest that for fruity vector esters, the effect of treatments was greater when the grapes were not fully ripe. Gómez-Plaza et al. [38] found, in Monastrell wines, no clear increase in fruity vector in wines made from MeJ-treated grapes, while in Tempranillo wines made from MeJ grapes, Garde-Cerdán et al. [17] found that behavior differed depending on the season.

#### 2.2.4. Terpenes

Terpenes are the most extensively studied compounds in many grapevine varieties, where they are responsible for varietal aromas, especially in aromatic grapes such as Muscat that may reach more than 5 mg/kg, while in neutral varieties this value may be a hundred times lower [1]. However, it has been reported that climate and management practices, such as the use of elicitors, can influence the final terpene concentration in wines [46]. Monastrell is a neutral variety with a poor terpene content [7], as can be seen from the findings reported in the present experiment.

Regarding seasons and treatments, in general terms, MeJ, but not Nano-MeJ, increased the terpene content of wines. Other authors, such as Yue et al. [47], also found a higher monoterpene content in Muscat Hamburg wines from grapes treated with MeJ.

More specifically, linalool, citronellol, and nerolidol were detected in 2019 and 2021 wines, the linalool concentration being much greater in 2019 (140%) in the MeJ treatment, while in 2021 a large increase was detected in linalool (80%) and citronellol (57%) concentrations in the MeJ treatment but not in the Nano-MeJ treatment. In contrast, in 2020, nerolidol was not detected in any treatment, while citronellol and linalool concentrations increased, but not significantly and only in wines elaborated with MeJ-treated grapes. Other authors, such as Gomez Plaza et al. [36] in regard to Monastrell wines and D´Onofrio et al. [32] in regard to Sangiovese wines, found an increment in linalool and citronellol when MeJ treatments were applied in grapes.

#### 2.2.5. Thioderivates

Regarding thioderivate volatile compounds, only 3-(methylthio)-1-propanol or methionol was detected. Its descriptor has been stated as a cooked cabbage odor, which is produced during alcoholic and malolactic fermentation by the catabolism of methionine [2]. Both treatments (MeJ and Nano-MeJ) increased the methionol concentration in 2019 and 2020, probably due to the increase in methionine in the grapes treated with MeJ as mentioned by other authors [14,29]; however, in 2021, neither treatment had any effect. Ruiz-García et al. [16] detected a substantial increase of this compound in Monastrell wine made from grapes treated with a combination of benzothiadiazole and methyl jasmonate, but Gutierrez-Gamboa et al. [46] found no differences in this compound in wines made from Tempranillo grapes treated with different doses of seaweed.

### 2.3. Multivariable Factorial Analysis

A multivariable factorial analysis (MFA) was made to determine which factors, among season and treatment, affected each volatile compound and whether there was any interaction between them. Table 3 shows the percentage of variance attributable to each of them. The MFA showed that season was a major factor for all the volatile compounds. The interaction between treatment and season was also an important factor of variation for some volatile compounds, whereas the treatment factor was only important as a factor of variation in a low fraction of the aromatic composition of the wines.

Season had a strongly significant effect on all the studied volatile compounds except Z-3-hexen-1-ol. The rest of the analyzed compounds were strongly influenced by the year, although some compounds, such as ethyl-decanoate, ethyl-acetate, and methionol, were much less affected. Among vintages, differences were due to climatic conditions (Figure 1) and the differences in the maturation stage of the berries at harvest time (Table 1).

Treatment factor had a strongly significant effect on most esters, except ethyl-acetate, which was not affected by the treatment, as well as on most terpenes, except nerolidol.

The interaction between treatment and season had a strongly significant effect (*p* < 0.001) on some esters, such as ethyl-decanoate, ethyl-dodecanoate, and diethyl-succinate, and one fatty acid, hexanoic acid. A slightly weaker interaction (*p* < 0.01 and *p* < 0.05) was observed for methionol, linalool, 3 methyl-1-butanol-acetate, and 3 methyl-1-butanol.

### 2.4. Multivariate Discriminant Analysis

A discriminant analysis was performed in order to check whether the measured variables in wines at the end of alcoholic fermentation during the three seasons could be used to classify our samples according to the applied treatments. This analysis is a tool that separates, by linear relation, a cluster of variables into groups and provides the possibility of classifying newly measured variables in the groups previously established.

Figure 2 shows that, two statistically significant (*p* < 0.05) canonic discriminating functions were obtained, explaining 100% of the variance, Function 1 being the most important. The relative percentage was 75.1% for Function 1 and 24.9% for Function 2. Both treatments were placed in the right part of the graph. The separation between the Nano-MeJ treatment and the control was not as great as between MeJ and the control, indicating greater differences with respect to control samples. However, MeJ and Nano-MeJ were close, indicating that, between them, the differences were not so evident, in spite of the 10-fold lower dose in the Nano-MeJ treatment. The standardized coefficient showed that 1-propanol, 3-methyl-1-butanol-acetate, 3-methyl-1-butanol, and ethyl-hexanoate contributed most to Function 1 and 3-methyl-1-butanol and β-phenyl-ethanol to Function 2. These results pointed to good separation among the treatments applied.

### 2.5. Sensory Analysis

Finally, in an attempt to confirm that the results of the analytical studies were reflected at the sensory level, triangular tasting was carried out to confirm whether there were differences between the control wines and the treatments. Figure 3 shows the results obtained in the triangular sensory analysis for the three seasons studied, in which only the correct responses by the tasters are represented when distinguishing between the samples that were different and those that were the same.

Specifically, Figure 3A shows a comparison of control wines vs. wines from the MeJ-treated grapes. As can be observed, during the first year, when the control wine was compared with the MeJ wine, 67% of tasters preferred the wine made from the MeJ-treated grapes; in 2020, 64% of tasters preferred the wine from MeJ-treated grapes; and in the last year, 88% of tasters preferred the MeJ wine. As can be observed from the results, the tasters always showed a preference for the wine that was made with grapes treated with MeJ, a preference that increased in the last year of the study.

Figure 3B shows a comparison of control wines vs. wines from Nano-MeJ-treated grapes. In the first year, the tasters preferred the wines from the Nano-MeJ batch. In the following years, this preference was 70% in 2020 and 75% in 2021.

These results show that the application of the treatments modified the wines sensorily, increasing their quality and the probability of greater acceptance on the part of consumers.

## 3. Materials and Methods

### 3.1. Chemicals

Sodium citrate tribasic dihydrate (Na_3_(C_6_H_5_O_7_)·2H_2_O, ≥99.0% pure) (Na_3_Cit), anhydrous potassium phosphate dibasic (K_2_HPO_4_, ≥99.0% pure), sodium carbonate (Na_2_CO_3_, ≥99.0% pure), calcium nitrate tetrahydrate (Ca(NO_3_)_2_·4H_2_O, ≥99.0% pure), and methyl jasmonate (C_13_H_20_O_3_, 95.0%, racemic), 1 propanol, 2-methyl-1-propanol, 3-methyl-1-butanol, 1-hexanol, E-3-hexen-1-ol, β-phenyl-ethanol, ethyl-acetate, 3-methyl-1-butanol-acetate, ethyl-caprate, ethyl-hexanoate, ethyl-octanoate, ethyl- dodecanoate, diethyl-succinate, ethyl-tetradecanoate, ethyl-hexadecanoate, nerolidol, citronellol, linalool, octanoic acid, hexanoic acid, 3-methyl-thio-1-propanol, 2-octanol and 4-methyl-2-pentanol were supplied by Sigma Aldrich (Madrid, Spain). NaCl, NaOH 0.1N, and ethanol 96% were from Panreac (Barcelona, Spain). Ultrapure water came from a Milli-Q system (Millipore Corp., Bedford, MA, USA). Tartaric acid from Laffort (Bordeaux, France) was used to adjust model wine. The TDI enzymatic kit was used for L-Malic acid analysis (Tecnología Difusion Ibérica S.L., Gavá, Spain)

### 3.2. Synthesis of Nanoparticles Doped with MeJ (Nano-MeJ)

A batch precipitation method at room temperature was used to synthesize calcium phosphate nanoparticles, following previously reported conditions [23]. Briefly, the method consisted of mixing two solutions (1:1 *v*/*v*): (A) Ca(NO_3_)_2_ (0.2 M) and Na_3_Cit (0.2 M) and (B) K_2_HPO_4_ (0.12 M) and Na_2_CO_3_ (0.1 M), stirring for 5 min. The precipitates were collected and washed with ultrapure water by centrifugation (5000 rpm, 15 min, 18 °C). Then, 200 mg of nanoparticles were vigorously mixed into 10 mL of ultrapure water with vortex and 20 mg of MeJ were added to the nanoparticle suspension. After 24 h stirring at room temperature, nanoparticles doped with MeJ (Nano-MeJ) was separated by centrifugation (12,000 rpm, 15 min, 18 °C) and stored at 4 °C.

### 3.3. Vegetal Material and Open Field Treatments

The treatments were carried out at an experimental field station, El Chaparral, located in Bullas (Murcia, Spain), latitude 38.11179 and longitude −1.6808, during three vintages 2019, 2020, and 2021. The vineyards were 17-year-old *Vitis vinifera* L. Monastrell variety on Richter 110 rootstock, trained in a bilateral cordon system. The plantation framework was 3 m × 0.8 m with a density of 4167 plants/ha.

The climate is semi-arid continental, the soil loamy sand. Figure 1A–C) shows the meteorological information related to monthly temperatures (°C) and rainfall (Lm^−2^) for each vintage, as recorded by the Agricultural Information System of the Region of Murcia (SIAM), located 50 m from the experimental field.

Three treatments were applied in triplicate, with 10 vines per replicate: i) control (water), ii) 10 mM methyl jasmonate (MeJ), and iii) nanoparticles doped with methyl jasmonate (Nano-MeJ) at 3.6 g L **^−^**^1^ (equivalent to 1mM in MeJ). The treatments were carried out using aqueous suspensions with Tween 80 (Sigma Aldrich, St. Louis, MO, USA) as the wetting agent (0.1% *v*/*v*). The leaves of each plant were sprayed with 200 mL of their corresponding treatment at veraison and a week later. A maturity control (°Brix, pH, and total acidity) was carried out to detect technological maturity and to establish harvest time. After manual harvest, the grapes were transported in 15 kg boxes to the winery located in the Oenological Station (Jumilla, Spain).

### 3.4. Vinifications

Each replicate (50 kg of grapes) was separately destemmed, crushed, and sulfited (0.08 g SO_2_/kg); total acidity was adjusted to 5.5 g L^−1^ with tartaric acid and inoculated by neutral aroma yeasts at 0.25 g L^−1^ (Zymaflore FX10, Laffort). All vinifications were made place in 50 L stainless tanks at a temperature of 23 ± 2 °C for 14 days, punching the cap down daily.

At the end of alcoholic fermentation (AF), each replicate was pressed, and the free-run and pressed wine were collected together in stainless steel 50 L tanks. After racking twice over two days, each replicate was stored in a bag in box. All sample wines were analyzed in triplicate at final alcoholic fermentation and in the same season that they were produced.

### 3.5. Physicochemical Analysis in Must and Wine

A must sample (after crushing) was collected to determine the following oenological parameters: total soluble solids (°Brix) using an Atago RX-5000X refractometer (Atago CO., LTD, Tokyo, Japan), pH, and titratable acidity with a Schott, alpha plus TA20 (SCHOTT-GERÄTE GmbH, Mainz, Germany) using a glass electrode (Xylem analytics Germany GmbH) and tartaric and malic acid with a CETLAB 600 (Microdom, Taverny, France).

The following parameters were analyzed in wines at the end of alcoholic fermentation: pH, titrable acidity, and malic acid, following the same methods as for the must. Volatile acidity was analyzed with a continuous flow QuAAtro analyzer (SEAL Analylical GmbH, Norderstedt, Germany) and the alcohol percentage was determinated (*v*/*v*) with an Anton Paar SP-1 MWine Alcolyzer (Anton Paar GmbH, Graz, Austria).

### 3.6. Isolation of Volatile Compounds in Wines by HS-SPME-GC-MS

Wine volatile composition was determined by headspace solid-phase micro extraction (HS-SPME) according to the methodology proposed by Moreno-Olivares et al. [48]. To extract the volatile compounds in wine samples, a divinylbezene-carboxene-polydimethylsiloxane fiber was used (Supelco, Bellefonte, PA, USA) and 3 g NaCl, 25µL of 2-octanol (100 µg L **^−^**^1^) and 25 µL of 4-metthyl-2-pentanol (2.5 mg L^−1^) as internal standards were added to ten milliliters of each wine sample in a 20 mL glass vial with a magnetic screw top and polytetrafluoroethylene (PTFE)-lined septum. A Gerstel auto-sampling tool (Gerstel GmbH&Co. KG, Mellinghofen, Germany) was used to condition the samples for 15 min at a controlled temperature of 40 °C with stirring. After concluding the extraction process, the analyses were performed on an HP 7890B gas chromatography (GC) coupled to an HP 5977A quadrupole mass spectrometer (Agilent Technologies, Palo Alto, CA, USA). Injections were carried out in splitless mode for 0.75 min, using a 2 mm inner diameter non-deactivated direct liner and the desorption time was 5 min. A DB-WAXetr capillary column (30 m × 250 µm, 0.25 μm film thickness; Agilent Technologies) was used for the chromatographic separation. The carrier gas was helium 8.0 (Abelló Linde SA, Barcelona Spain) with a column head pressure of 8 psi. The oven temperature was set at 40 °C for 0.5 min, raised to 260 °C at 4 °C min^−1^ and then held at that temperature for 5 min. The mass spectrometer detector was operated in electron ionization mode at 70 eV and in scan mode (mass range 20–350 amu) and the transfer line to the mass spectrometer system was maintained at 230 °C. Peaks were identified with the mass library (Wiley Registry 6.0; Wiley, Chichester, UK). Quantitative analysis was performed by total ion current using the calibration curves obtained for each compound, which were prepared using a synthetic wine model (3 g tartaric acid, 12% ethanol, and pH adjusted to 3.5 with NaOH).

### 3.7. Sensory Analysis

The sensory analysis was performed by 12 experienced tasters, all members of the Oenological Station of Jumilla. Wines were subject to a sensory triangular test in order to check whether there were detectable differences between the wines made with the differently treated grapes and the control wine. The tasting was organized as follows: three samples (40 mL each) were presented in clear official glasses in coded random order, two of which were identical and the other different. Each taster had to select the sample which they considered different (forced election) and then to indicate which sample they preferred. The statistical significance of the number of correct judgements versus the number of judgements was subsequently determined according to the AENOR norm (ISO 4120:2021).

### 3.8. Statistical Analysis

The results are shown as an average of three replicates for each measured parameter Significant differences among treatments for each variable were assessed by analysis of variance (ANOVA) using RStudio 3.6.2 (Boston, MA, USA), The LSD test was used to compare the means (*p* < 0.05) when the ANOVA test was significant. In addition, a multivariate factorial analysis and a linear discriminant analysis were performed, using Statgraphics Centurion XVIII.

## 4. Conclusions

This study reports on the volatile composition of wines from treated (MeJ and Nano-MeJ) and untreated grapes, with the aim of establishing the influence of applications of these treatments on the final composition of the wines.

Season was the most important factor that affected the development of volatile compounds during the three studied years. The lowest volatile content was obtained in the first year because of the high rainfall before harvest. Regarding the treatments, both MeJ and Nano-MeJ produced a substantial increase in the concentration of fruity esters, although these increases were greater when grapes were less ripe. With respect to other families of compounds, higher alcohols were enhanced in 2020 and 2021, and fatty acids in 2021. Comparing both treatments, the MeJ treatment increased the terpene fraction more than the Nano-MeJ treatment, which at the applied concentration had no effect on this fraction. In addition, the results obtained in the sensory analysis showed a similar preference for MeJ and Nano-MeJ wines, compared with the control. However, it should be noted that the Nano-MeJ effect was obtained with a ten times lower dose of MeJ than that usually applied. Therefore, the use of nanoparticles doped with MeJ can be considered an interesting alternative to conventional MeJ treatments for obtaining aromatic high-quality wines; the reduced dosage of agrochemicals contributes to a more sustainable and efficient viticulture.

## Figures and Tables

**Figure 1 molecules-27-02878-f001:**
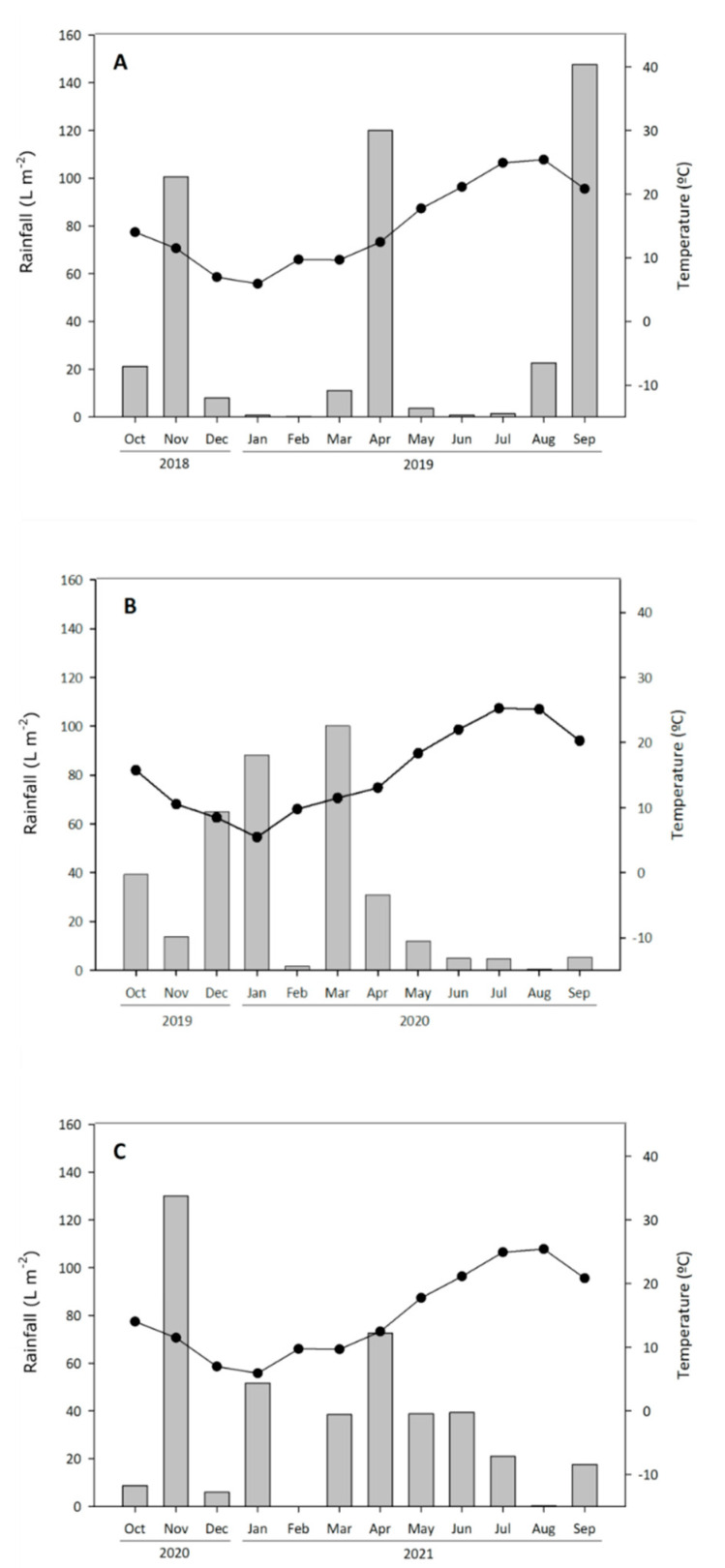
Monthly rainfall (bars) and average temperature (dots) for the three seasons studied. (**A**) 2019, (**B**) 2020, and (**C**) 2021.

**Figure 2 molecules-27-02878-f002:**
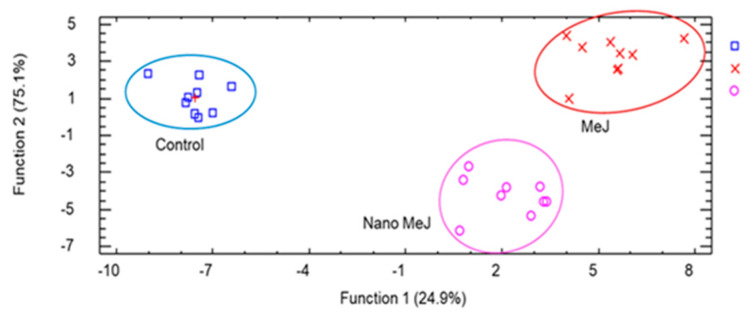
Canonical discriminant analysis used to classify the wines by treatments.

**Figure 3 molecules-27-02878-f003:**
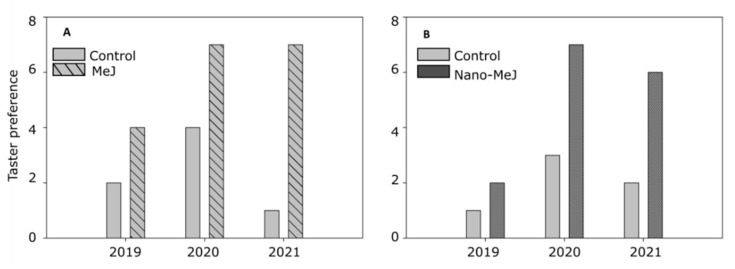
Triangular sensorial analysis of wines. (**A**) Wines from untreated grapes (control) vs. grapes treated with MeJ. (**B**) Wines from untreated (control) vs. grapes treated with Nano-MeJ. Bars show the number of tasters who preferred each wine.

**Table 1 molecules-27-02878-t001:** Oenological parameters in Monastrell musts and wines elaborated from untreated (control) and treated grapes with foliar applications of Methyl Jasmonate (MeJ) and Nano Methyl Jasmonate (Nano-MeJ) at the end of alcoholic fermentation (AF) in 2019, 2020, and 2021 seasons.

	2019				2020				2021			
	Control	MeJ	Nano-MeJ	*p*-Value	Control	MeJ	Nano-MeJ	*p*-Value	Control	MeJ	Nano-MeJ	*p*-Value
**MUST**												
°Brix	23.13 ± 0.75	22.90 ± 0.40	23.17 ± 0.76	ns	25.33 ± 0.6	25.57 ± 0.7	24.87 ± 0.7	ns	23.55 ± 0.38	23.82 ± 0.28	23.80 ± 0.30	ns
Total acidity	2.85 ± 0.21	3.19 ± 0.16	2.91 ± 0.06	ns	2.34 ± 0.09 b	2.68 ± 0.19 a	2.39 ± 0.11 b	*	1.92 ± 0.13 b	2.39 ± 0.11 a	2.03 ± 0.05 b	**
pH	3.88 ± 0.03	3.84 ± 0.03	3.86 ± 0.05	ns	4.09 ± 0.02	4.09 ± 0.04	4.14 ± 0.02	ns	4.11 ± 0.06 a	4.00 ± 0.04 b	4.05 ± 0.02 ab	*
Tartaric acid	3.90 ± 0.13	3.80 ± 0.07	3.89 ± 0.08	ns	4.49 ± 0.15	4.41 ± 0.15	4.21 ± 0.04	ns	3.97 ± 0.08	3.91 ± 0.05	3.86 ± 0.16	ns
Malic acid	1.41 ± 0.35	1.66 ± 0.19	1.44 ± 0.10	ns	1.44 ± 0.06b	1.89 ± 0.04 a	1.45 ± 0.04 b	***	1.04 ± 0.07 b	1.41 ± 0.12 a	1.16 ± 0.02 b	**
**WINE AF**												
Alcohol %	12.94 ± 0.16	12.95 ± 0.37	13.37 ± 0.45	ns	14.72 ± 0.34	14.45 ± 0.09	14.40 ± 0.52	ns	13.44 ± 0.15	13.56 ± 0.33	13.86 ± 0.10	ns
Volatile acidity	0.40 ± 0.04	0.41 ± 0.06	0.38 ± 0.02	ns	0.41 ± 0.04 ab	0.44 ± 0.03 a	0.36 ± 0.01 b	*	0.40 ± 0.02 a	0.40 ± 0.05 a	0.28 ± 0.01 b	**
Total acidity	7.44 ± 0.11 a	6.62 ± 0.15 b	6.75 ± 0.07 b	**	7. 04 ± 0.16	6.45 ± 1.12	6.42 ± 0.14	ns	7.05 ± 0.06 a	6.49 ± 0.11 c	6.77 ± 0.20 b	**
pH	3.33 ± 0.08	3.38 ± 0.02	3.41 ± 0.02	ns	3.43 ± 0.02	3.59 ± 0.24	3.48 ± 0.02	ns	3.38 ± 0.03 b	3.46 ± 0.04 a	3.39 ± 0.02 b	*
Malic acid	1.69 ± 0.32	1.89 ± 0.17	1.73 ± 0.02	ns	1.44 ± 0.04 b	1.69 ± 0.10 a	1.51 ± 0.03 b	**	1.35 ± 0.07 b	1.56 ± 0.07 a	1.44 ± 0.05 ab	*

Tartaric acid, malic acid, volatile acidity (as acetic acid), and total acidity (as tartaric acid) are expressed in g L^−1^. All the parameters are given with their standard deviation (n = 3). For each parameter and season, different lowercase letters indicate significant differences between treatments (*p ≤* 0.05). Statistically significant at * *p ≤* 0.05, ** *p ≤* 0.01 and *** *p ≤* 0.001, respectively. ns: *not significant.*

**Table 2 molecules-27-02878-t002:** Volatile compounds (mg L*^−^*^1^) in Monastrell wines elaborated from untreated grapes (control) and grapes treated with Methyl Jasmonate (MeJ) and Nano Methyl Jasmonate (Nano-MeJ) in 2019, 2020, and 2021 seasons.

		2019			2020			2021		Multifactorial analysis		
**Volatile compounds**	**Control**	**MeJ**	**Nano-MeJ**	**Control**	**MeJ**	**Nano-MeJ**	**Control**	**MeJ**	**Nano-MeJ**	**S**	**T**	**SxT**
**FATTY ACIDS**												
Hexanoic acid	nd	nd	nd	5.62 ±0.08ab	6.57±0.45 a	4.73±0.60 ab	3.92±0.12 b	4.90±0.50 a	5.01±0.10 a	***	**	**
Octanoic acid	0.891±0.029	1.01±0.00	1.05±0.12	3.78±1.17	4.40±1.61	4.14±0.67	3.54 ±0.34 b	4.40±0.56 a	4.79±0.11 a	***	ns	ns
Decanoic acid	nd	nd	nd	nd	nd	nd	0.12±0.02	0.14±0.01	0.15±0.01	***	ns	ns
**Total fatty acids**	**0.89 ±0.029**	**1.01±0.00**	**1.05±0.12**	**9.40±2.00**	**10.97±1.29**	**8.87 ±0.25**	**7.58±0.40 b**	**9.44±1.07 a**	**9.95±0.22 a**	*******	*****	*****
**ALCOHOLS**												
1-propanol	11.83±0.50 b	16.18±2.77 a	16.51±1.91 a	54.28±10.1	50.30±0.66	49.40±4.32	18.29±3.94	18.53±2.53	19.20±2.15	***	ns	ns
1-butanol	nd	nd	nd	nd	nd	nd	3.73±1.32	4.77±1.03	4.36±1.30	***	ns	ns
2-methyl-1-propanol	61.42±3.68	70.22±5.22	69.53±6.03	101.59±3.06	101.37±13.96	95.93±4.09	77.60±4.39	87.57±8.53	90.98±3.77	***	ns	ns
2-methyl-1-butanol	80.92±7.91	96.73±8.28	96.54±7.36	152.87±8.02	146.06±6.54	141.07±5.91	106.99±6.38	112.01±15.12	117.77±2.93	***	ns	ns
3-methyl-1-butanol	150.28±14.68	179.65±15.37	179.29±13.66	283.91±14.9	271.25±12.14	261.99±10.98	264.32±20.40	301.69±18.54	294.63±10.34	***	ns	*
1-hexanol	4.27±0.56	4.73±0.37	4.27±0.25	4.67±0.16 b	5.03±0.18 a	5.13±0.12 a	3.51±0.06 b	4.03±0.24 a	3.72±0.23 ab	***	*	ns
Z-3-hexen-1-ol	nd	nd	nd	nd	nd	nd	0.005 ±0.006	0.198 ±0.32	0.02 ±0.009	ns	ns	ns
E-3-hexen-1-ol	0.311±0.023	0.365±0.037	0.303±0.016	0.303±0.027	0.346±0.016	0.314±0.035	nd	nd	nd	***	**	ns
β-phenyl-ethanol	49.74±2.56 b	59.38±1.44 a	60.60±3.51 a	103.88±16.79	102.16±5.54	95.95±6.19	60.46±2.90	68.75±9.98	71.83±4.19	***	ns	ns
**Total alcohols**	**358.77±26.86**	**427.26±33.34**	**427.03±32.29**	**701.52±48.00**	**676.51±29.11**	**649.79±28.99**	**534.91±30.35**	**597.54±51.70**	**602.50±8.19**	*******	**ns**	*****
**ESTERS**												
Ethyl acetate	46.09±10.91	48.79±6.32	37.92±4.26	56.52±21.58	69.47±5.55	71.39±4.88	67.93±10.41	67.21±7.07	60.68±6.03	***	ns	ns
3-methyl-1-butanol-acetate	0.597±0.185 b	1.40±0.20 a	1.28±0.16 a	3.92±0.13 c	5.88±0.36 a	4.44±0.22 b	1.055±0.134 b	2.45±0.61 a	1.99±0.04 a	***	***	**
Ethyl-hexanoate	0.229±0.076 b	0.440±0.107 a	0.358±0.036 ab	1.19±0.08 b	1.61±0.13 a	1.46±0.07 a	0.596±0.029 b	1.026±0.148 a	0.912±0.084 a	***	***	ns
Ethyl-octanoate	1.07±0.07 b	1.96±0.39 a	1.67±0.11 a	4.63±0.65	5.90±0.41	5.63±0.51	2.433±0.188 b	3.916±0.586 a	3.807±0.419 a	***	***	ns
Ethyl-decanoate	0.067±0.011	0.069±0.007	0.057±0.005	0.255±0.072 b	0.339±0.051 b	0.528±0.109 a	0.111±0.021 b	0.179±0.028 a	0.151±0.023 ab	***	**	***
Ethyl-dodecanoate	0.136±0.000	0.137±0.000	0.136±0.000	0.159±0.009 b	0.175±0.023 b	0.251±0.044 a	0.140±0.002 b	0.147±0.002 a	0.144±0.001 a	***	**	***
Ethyl-tetradecanoate	0.014±0.003	0.019±0.003	0.016±0.005	0.043±0.016	0.065±0.021	0.083±0.021	0.014±0.002 c	0.030±0.003 a	0.024±0.003 b	***	*	ns
Ethyl-hexadecanoate	0.008±0.001	0.011±0.001	0.011±0.002	0.050±0.020	0.059±0.010	0.074±0.014	0.011±0.002 b	0.032±0.006 a	0.032±0.004 a	***	**	ns
diethyl succinate	0.036±0.000 b	0.300±0.027 a	0.270±0.021 a	0.593±0.061	0.593±0.079	0.575±0.036	0.519±0.010 c	0.659±0.088 b	0.809±0.055 a	***	***	***
**Total esters**	**48.25±11.07**	**53.13±6.92**	**41.72±4.52**	**67.36±20.67**	**84.09±5.93**	**84.43±4.41**	**72.80±10.17**	**75.65±8.36**	**68548±6.28**	*******	**ns**	**ns**
**TERPENES**												
Linalool	0.005±0.003 b	0.012±0.002 a	0.005±0.000 b	0.008±0.003	0.012±0.002	0.007±0.000	0.015±0.001 b	0.027±0.002 a	0.015±0.000 b	***	***	**
citronellol	0.016±0.002	0.019±0.001	0.017±0.002	0.025±0.002 ab	0.030±0.004 a	0.022±0.002 b	0.014±0.000 b	0.022±0.003 a	0.016±0.000 b	***	***	ns
Nerolidol	0.006±0.001	0.007±0.001	0.006 ±0.000	nd	nd	nd	0.006±0.000 b	0.007±0.000 a	0.007±0.00 a	***	ns	ns
**Total terpenes**	**0.027±0.002 b**	**0.038±0.003 a**	**0.028±0.003 b**	**0.033±0.005 b**	**0.042±0.002 a**	**0.029±0.002 b**	**0.035±0.001 b**	**0.055±0.004 a**	**0.037±0.000 b**	*******	*******	*****
**MISCELANEOUS**												
3-(methylthio)-1-propanol	3.79±0.49 b	5.92±0.13 a	5.78±0.35 a	5.67±0.39 b	7.00±0.34 a	6.23±0.58 ab	4.31±0.73	4.52±0.51	4.16±0.29	***	***	**
**TOTAL**	**411.73±34.92**	**487.35±39.61**	**475.60±36.93**	**778.32±29.52**	**771.60±34.25**	**743.12±31.78**	**619.64±40.68**	**687.20±61.29**	**685.20±14.52**	*******	*****	**ns**

All the parameters are given with their standard deviation (n = 3). For each parameter and season, different lowercase letters indicate significant differences between treatments (*p* ≤ 0.05). nd: not detected; S: Season; T: Treatment; SxT: interaction between season and treatment factors. Statistically significant at * *p* ≤ 0.05, ** *p* ≤ 0.01 and *** *p* ≤ 0.001, respectively. ns: not significant.

**Table 3 molecules-27-02878-t003:** Percentage of variance attributable to treatment (T), season (S) and interaction of the variables for each volatile compound concentration in Monastrell wines.

Volatile Compound	T (%)	S (%)	TxS (%)	Error (%)
Hexanoic acid	1.30 **	94.13 ***	2.87 ***	1.70
Octanoic acid	2.57 ns	82.64 ***	1.61 ns	13.19
Decanoic acid	0.28 ns	97.80 ***	0.57 ns	1.35
**Total fatty acids**	**0.67 ***	**96.67 *****	**1.20 ***	**1.45**
1-propanol	0.00 ns	94.76 ***	1.08 ns	4.15
1-butanol	0.45 ns	91.21 ***	0.91 ns	7.43
2-methyl-1-propanol	3.28 ns	78.63 ***	4.83 ns	13.25
2-methyl-1-butanol	0.85 ns	87.03 ***	4.61 ns	7.51
3-methyl-1-butanol	1.87 ns	89.31 ***	3.97 *	4.86
1-hexanol	9.97 *	71.29 ***	3.24 ns	15.49
E-3-hexen-1-ol	0.94 **	97.12 ***	0.58 ns	1.35
Z-3-hexen-1-ol	7.51 ns	11.30 ns	15.02 ns	66.17
β-phenyl-etanol	1.42 ns	86.28 ***	3.32 ns	8.98
**Total higher alcohols**	**1.64 ns**	**88.81 *****	**4.07 ***	**5.49**
Ethyl acetate	2.98 ns	52.41 ***	10.16 ns	34.45
3-methyl-1-butanol-acetate	10.92 ***	85.19 ***	2.16 **	1.73
Ethyl-hexanoate	9.67 ***	86.90 ***	0.84 ns	2.59
Ethyl-octanoate	9.77 ***	85.48 ***	0.66 ns	4.09
Ethyl-decanoate	7.35 **	73.34 ***	12.45 ***	6.87
Ethyl-dodecanoate	13.01 **	48.51 ***	25.25 ***	13.23
Ethyl-tetradecanoate	8.98 *	69.15 ***	7.58 ns	14.29
Ethyl-hexadecanoate	7.45 **	79.16 ***	3.58 ns	9.81
diethyl succinate	10.29 ***	78.59 ***	7.79 ***	3.33
**Total esters**	**4.34 ns**	**64.78 *****	**7.03 ns**	**23.85**
Linalool	28.08 ***	61.98 ***	5.34 **	4.60
Citronellol	24.59 ***	58.39 ***	6.07 ns	10.95
Nerolidol	0.44 ns	97.68 ***	0.68 ns	1.20
**Total terpenes**	**54.47 *****	**31.37 *****	**7.04 ***	**7.12**
3-(methylthio)-1-propanol	21.23 ***	53.80 ***	13.64 **	11.33

Statistically significant at * *p* ≤ 0.05, ** *p* ≤ 0.01 and *** *p* ≤ 0.001, respectively. ns: not significant.

## Data Availability

Data is contained within the article.

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
