# Peer review of "Effects of Methyl Jasmonate and Nano-Methyl Jasmonate Treatments on Monastrell Wine Volatile Composition"

_molecules, 2022, doi:10.3390/molecules27092878_

Round 1

Reviewer 1 Report

The article by Gimenez-Banon, is timely and relevant to the field of food chemistry. I would be willing to accept the manuscript provided that the authors address the following issues.

The results and discussion should come after the materials and methods section. How was the pH meter verified, the same can be said about the GC-MS, there is no discussion on whether these two instruments were in good operating condition, when the analysis was done. There has been usage of multivariate techniques done in the past to separate out wines and spirts, the authors should cite the following paper: Burns, Rachel L., et al. "A Fast, Straightforward and Inexpensive Method for the Authentication of Baijiu Spirit Samples by Fluorescence Spectroscopy." Beverages 7.3 (2021): 65.

I believe this is good work, however the authors need to address these concerns.

Author Response

Reviewer 1

List of changes in the current main document / or a rebuttal

The results and discussion should come after the materials and methods section

How was de pH meter verified

GC-MS verification

There has been usage of multivariate techniques done in the past to separate out wines and spirits, the authors should cite the following paper: Burns, Rachel L., et al. “ A fast, straightforward and inexpensive method for the authenticacion of Baiju Spirit Samples by Fluorescence Spectrospocopy

Our sincere thanks for the review of the article, and for the opinions generated by the reviewer. This journal has a template to present the paper and we have followed the instructions given by the journal to the authors. This is the reason why we have written the different sections of the article in this order.

We have not detailed the calibration mode of the pH meter due to the fact of it is a basic operation that we perform every day in our lab before any series of measurements, and we considered that was not necessary to specify , either way we calibrate it following the instrument instructions and using two certified standards at pH 4 and pH 7

The equipment is verified daily in our laboratory before analyzing each batch of samples by performing an e-tune. This is an operation to verify that all the parameters of the equipment are correct. Once this is done, we begin to analyze our samples.

We appreciate the suggested article, we have reviewed it and it is interesting the use of emission spectra to differentiate different types of liquor samples. But we do not see a direct relationship with the determination of the aromatic profile of the wines. The used of discriminant analysis to separate groups is very common and because it is a statistical tool we didn´t consider necessary to add any reference.

Reviewer 2 Report

The authors investigated the influence of methyl jasmonate treatments on monastrell wine volatile composition.  The experimental design and analytical approach were appropriate regarding the main aim of this study. The manuscript is well written and easy to follow.

I just have some major and a couple of minor comments.

Please carefully reread the manuscript, there are many typo mistakes. Here are some:

L39, 100, 115-119, 185, 212, 221, 234, 246, 248, 265, 267, 294, 319, 335 (Fig 2B), 397-399, 411-413, 425, 433,

Please write the numbers of units in superscript and the numbers of chemical formula in subscript.

L425: replace ppb and ppm by µg/L and mg/L

Have the wines been analyzed over 3 years (meaning wines made in 2019 analyzed in 2019, wines from 2020 analyzed in 2020…) or have they been analyzed at the same time (all three wines analyzed in 2021)? If the wines were analyzed at the same time this could also explain why the lowest volatile content was observed in the first year wine.

Same question for sensory analysis. Did the authors performed sensory analysis over the three years of the study or at the end? If they performed tests over 3 years, did they have the same judges?

Concerning volatile compound analyses by HS-SPME-GC-MS, why did the authors choose scan mode instead of SIM mode? SIM mode is much more appropriate for quantitation than SCAN mode.

Moreover, the authors prepared calibration solutions in model wine. Did the authors investigate a potential matrix effect? Is it possible that the quantification is overestimated with a calibration curve prepared in model wine?

Concerning sensory analysis, please provide results of statistical analysis of triangular tests. The preference tests require a large number of tasters, which is not the case in this study since there are only 12 tasters. The statistical results on taster preference do not really make sense here with only 12 persons, and therefore I suggest removing these results and only discussing the results of the triangular tests.

Statistical analyses: the authors used one-way ANOVA to analyze the data. Did they verify if the data was normally distributed? Please provide results of normality tests.

Author Response

Reviewer 2

Please carefully reread the manuscript, there are many tipo mistakes.

Please write the numbers of units in superscript an d the numbers of chemical formula in subscript

L425: replace ppb and ppm by µg/L and mg/L

Have the wines been analyzed over 3 years (meaning wines made in 2019 analyzed in 2019, wines from 2020 analyzed in 2020…) or have they been analyzed at the same time (all three wines analyzed in 2021)?

If the wines were analyzed at the same time this could also explain why the lowest volatile content was observed in the first year wine.

Same question for sensory analysis

Concerning volatile compound analyses by HS-SPME-GC-MS, why did the authors choose san mode instead of SIM mode? SIM mode is much more appropriate for quantitation than SCAN mode.

Moreover, the authors prepared calibration solutions in model wine. Did the authors investigate a potential matrix effect? Is it possible that the quantification is overestimated with a calibration curve prepared in model wine?

Concerning sensory analysis, please provide results of statistical analysis of triangular test. The preference test require a large number of tasters, which is not the case in this study since there are only 12 tasters. The statistical results on taster preference do not really make sense here with only 12 persons, and therefore I suggest removing the results and only discussing the results of the triangular test.

Statistical analyses: the authors used one-way ANOVA to analyze the data. Did they verify if the data was normally distributed? Please provide results of normality tests.

The errors mentioned in addition to others have been fixed through the manuscript in the following lines:

L15, L39,L52,L59,L63,L100,L113,L116,L119,L177,L183-185,L191,L207,L212,L216,L-220-221, L223,L234,L248,L265,L267,L294,L297,L311-314,L319,L335,L337,L352-356,L368-369,L376,L392,L397,L399,L433,L437,L441,L456

We have changed it in the manuscript.

The different wines were analyzed over 3 years for them to prevent evolution processes that would mask the effect of the treatments. So that question can be better  understood we have rewrite L402/403 :” All samples wine were analyzed by triplicate at final alcoholic fermentation and in the same season that them were produced”

If this were the case it would be probable but because every wine was analyzed at its production year and close to time of having finished the alcoholic fermentation, the probable reason on this difference was the season, as we said, 2019 register an important rainfalls before harvest time.

The sensory analysis was performed over the three years. The judges were the same, the workers of Oenological Station of Jumilla trained previously.

As the reviewer suggests, the SIM method is more appropriate to SCAN once the compounds of interest have been identified, but our methodology is that used by Moreno-Olivares et al. 2020, which in turn is based on that of Gomez-Plaza et al. 2012, which used the SCAN mode. Since our analyses are already done, we keep it in mind for the future.

As the reviewer suggests, this overestimation could occur, but the interest of our work was to compare the aromatic composition in wines in which different treatments had been applied. Therefore, this overestimation, if it were to occur, would not influence us in the comparison, since in all cases we are using the same matrix.

Regarding statistical analysis of triangular test, Roessler et al. 1 provides tables which indicate the number of tasters and the number of correct answers  to each significance level. The minimum number of tasters is 5.

As for preference test we admit that 12 tasters would be not enough to considerer statistical results. We have removed from the text these statistical consideration.

1Roessler, EB;Pangborn, RM;Sidel, JL; Stone H. Expanded statistical tables for estimating significance in paired-preference, paired-difference, duo-trio and triangle tests. J Food Sci 43:940–943 (1978).

We are verified the normality distribution of the data using Shapiro wilks test because our parameters has less of 50 data. We show some examples of it, because showing all the results would be too long.

2019

Normality

Tests for Normality for 1_propanol

Shapiro-Wilks W statistic = 0,904494

Yes

P-Value = 0,274541

Tests for Normality for 2_methyl_1_propanol

Shapiro-Wilks W statistic = 0,92666

 P-Value = 0,444019

Yes

Tests for Normality for 2_methyl_1_butanol

Shapiro-Wilks W statistic = 0,955852

Yes

P-Value = 0,747859

Tests for Normality for 3_methyl _1 butanol

Shapiro-Wilks W statistic = 0,955852

Yes

P-Value = 0,747859

Tests for Normality for 1_hexanol

Shapiro-Wilks W statistic = 0,986385

Yes

P-Value = 0,986942

Tests for Normality for E_3_hexen_1_ol

Shapiro-Wilks W statistic = 0,855256

Yes

P-Value = 0,0841374

Tests for Normality for _β_phenyl_etanol

Shapiro-Wilks W statistic = 0,936751

Yes

P-Value = 0,541636

Tests for Normality for Ethyl acetate

Shapiro-Wilks W statistic = 0,875705

Yes

P-Value = 0,139158

Tests for Normality for 3_methyl_1_butanol_acetate

Shapiro-Wilks W statistic = 0,920115

Yes

P-Value = 0,387359

Tests for Normality for Ethyl_hexanoate

Shapiro-Wilks W statistic = 0,937899

Yes

P-Value = 0,553438

Tests for Normality for Ethyl_octanoate

Shapiro-Wilks W statistic = 0,927972

Yes

P-Value = 0,456027

Tests for Normality for Ethyl decanoate

Shapiro-Wilks W statistic = 0,941142

Yes

P-Value = 0,587424

Tests for Normality for Ethyl dodecanoate

Shapiro-Wilks W statistic = 0,917369

Yes

P-Value = 0,365244

Reviewer 3 Report

This manuscript includes the analysis of the oenological parameters, the volatile composition, the multifactorial analysis, the canonical discriminant analysis, and the triangular sensorial analysis of Monastrell wines treated with nanopar-ticles doped with MeJ over three sea-sons in comparison with the application of MeJ in conventional way. This manuscript covers very interesting research and results, but this paper needs to be revised in order to be published in Molecules.

1. Check the words, symbols, and spaces in the text and results.
2. The text words presented in Fig. 2-3 are not clearly visible. At least, it needs to be modified to a similar size so that it can look like Fig.1.
3. The amount of total esters in the grape wine treated with Nano-MeJ was rather decreased compared to the control in 2019 and 2021. Even considering the error value of the control, the average value of the Nano-MeJ treatment group was low. What do you think is the reason?
4. Do you think the treatment technology using these nanoparticles will have an effect on the quality improvement of other fruit fermented alcoholic beverages?
5. Although calcium phosphate nanoparticles have various advantages, there are articles reporting that this induces apoptosis (Masouleh MP et al. Curr Pharm Des. 2017;23(20):2930-2951; Liu et al. J Mater Chem B. 2014 Jun 14;2(22):3480-3489). Can you comment on this?

Author Response

Reviewer 3

1. Check the words, symbols and spaces in the text and results.

2.The text words presented in fig. 2-3 are not clearly visible. At least, it needs to be modified to a similar size so that it can look like fig. 1

3. The amount of total esters in the grape wine treated with Nano- MeJ was rather decreased compared to the control in 2019 and 2021 Even considering the error value of the control, the average value of the Nano-MeJ treatment group was low. What do you think is the reason?

4. Do you think the treatment technology using these nanoparticles will have an effect on the quality improvement of other fruit fermented alcoholic beverages?

5. Although calcium phosphate nanoparticles have various advantages, there are articles reporting that this induces apoptosis. Can you comment on this?

We have done it and corrected them in the following lines in the manuscript:

L15, L39,L52,L59,L63,L100,L113,L116,L119,L177,L183-185,L191,L207,L212,L216,L-220-221, L223,L234,L248,L265,L267,L294,L297,L311-314,L319,L335,L337,L352-356,L368-369,L376,L392,L397,L399,L433,L437,L441,L456

We have replace for another figures with a bigger typography  in the manuscript.

The decrease in total esters is due to the fact that the ethyl acetate decrease, if we consider de fruity esters (total amount- ethyl acetate contribution) we observed and increased all the treatments vs. control and all the seasons.

We think that because MeJ induce the secondary metabolism in plants, its effects would be found in others fruits and would be transferred to other fermented beverages made with them, in addition as can be observed in our study season is an important factor to be considerered.

We have read the comment article and we think that is an important factor to considerer but in our work we have had no notice about apoptosis signals in the treated vines so we have no considered this fact.

Round 2

Reviewer 1 Report

One key function of a good analytical lab is to include QA/QC, without this I can't accept this manuscript. In my judgement any analytical instrument is routine, that includes GC-MS, SEM, ICP-MS, FTIR, Raman, Fluorescence, etc and pH meters. Many government laboratories must include how the pH meter was verified. Without this information, I can't accept the manuscript, as how do I know the standards to calibrate this instrument weren't expired? Should we accept data from expired standards?